# PPARγ/mTOR Regulates the Synthesis and Release of Prostaglandins in Ovine Trophoblast Cells in Early Pregnancy

**DOI:** 10.3390/vetsci9110649

**Published:** 2022-11-21

**Authors:** Kexing Hao, Jing Wang, Zhiyuan Li, Huihui Chen, Bin Jia, Guangdong Hu

**Affiliations:** College of Animal Science and Technology, Shihezi University, Shihezi 832000, China

**Keywords:** trophoblast cells, prostaglandins, PPARγ, mTOR, PGE_2_/PGF_2α_

## Abstract

**Simple Summary:**

During early gestation, a series of prostaglandins are synthesized and secreted by the sheep’s conceptus. The ratio of PGE_2_ to PGF_2α_ is very important for embryo attachment. In this process, the expression level of PPARγ also begins to increase, and prostaglandin metabolites are natural endogenous ligands of PPARγ. Inhibition of PPARγ activity can enhance mTOR signaling. However, the precise roles of PPARγ/mTOR during this process are unknown. Here, we found that inhibiting PPARγ activity in trophoblastic cells disequilibrated their synthesis and secretion of prostaglandins, while blocking the mTOR pathway restored it to normal levels. Our results demonstrate that PPARγ/mTOR are likely to function in the synthesis of prostaglandins. These findings extend our understanding and may provide new insights into the mechanism of PPARγ/mTOR regulation of prostaglandin secretion in trophoblastic cells.

**Abstract:**

Trophoblast cells synthesize and secrete prostaglandins (PGs), which are essential for ruminants in early gestation to recognize pregnancy. Hormones in the intrauterine environment play an important role in regulating PGs synthesis during implantation, but the underlying mechanism remains unclear. In this study, co-treatment of sheep trophoblast cells (STCs) with progesterone (P_4_), estradiol (E_2_), and interferon-tau (IFN-τ) increased the ratio of prostaglandin E2 (PGE_2_) to prostaglandin F2α (PGF_2α_) and upregulated peroxisome proliferator-activated receptor γ (PPARγ) expression, while inhibiting the mechanistic target of rapamycin (mTOR) pathway and activating cellular autophagy. Under hormone treatment, inhibition of PPARγ activity decreased the ratio of PGE_2_/PGF2α and cellular activity, while activating expression of the mTOR downstream marker—the phosphorylation of p70S6K (p-p70S6K). We also found that the PPARγ/mTOR pathway played an important role in regulating trophoblast cell function. Inhibition of the mTOR pathway by rapamycin increased the ratio of PGE_2_/PGF_2α_ and decreased the expression of apoptosis-related proteins after inhibiting PPARγ activity. In conclusion, our findings provide new insights into the molecular mechanism of prostaglandin regulation of trophoblast cells in sheep during early pregnancy, indicating that the PPARγ/mTOR pathway plays an important role in PGs secretion and cell viability.

## 1. Introduction

Embryo implantation is a process in which the blastocyst becomes attached to the endometrium through the gradual histological and physiological connection between the blastocyst’s trophoblast and the endometrium. Failure of embryo implantation is the main cause of loss of embryos produced in vitro [1]. Under natural conditions, sheep embryos enter the uterus on the 4th day of morula development and develop into blastocysts by the 6th day. During the blastocyst stage, embryos mature and have the ability to attach. After that, the blastocysts separate from the zona pellucida (day 8), extend to form filaments (days 11–16) and, finally, attach to the endometrium (day 16) [2]. Interferon-tau (IFN-τ) is an important pregnancy recognition signal secreted by the conceptus, which mainly acts on the endometrium, inhibiting upregulation of the oxytocin receptor and the pulsed release of prostaglandin F2α (PGF_2α_), so as to maintain the corpus luteum and promote embryo implantation [3,4]. IFN-τ mRNA expression can be detected in the ovine conceptus from days 10 to 11 of pregnancy recognition, with the highest expression by day 13 and decreasing through day 17 [5]. In the process of implantation, intrauterine function is stimulated and maintained by maternal progesterone (P_4_) and estradiol (E_2_), along with fetal IFN-τ [6]. 

Peroxisome proliferator-activated receptor γ (PPARγ)—a member of the nuclear receptor superfamily—is a ligand-activated transcription factor that regulates biological processes in a variety of tissues, including the placenta [7]. PPARγ plays a key regulatory role in lipid metabolism and promotes adipocyte differentiation [8]. It is highly expressed in the uteri of pigs, placentas of dogs, and placental cotyledons of ewes during the implantation period of pregnancy [9]. Transcriptome sequencing of bovine embryos showed that PPARγ transcription levels increased significantly at the beginning of pregnancy elongation and were expressed at high levels throughout conceptus elongation [10]. From days 12 to 17 after fertilization, the transcription level of PPARγ in placental trophoblast cells increased steadily, and trophoblast cell proliferation and lipid metabolism were observed [11]. Inhibition of PPARγ transcription resulted in severe developmental retardation of ovine concepti [12].

The self-degradation process known as autophagy is the main intracellular catabolic mechanism for degrading and recycling proteins and organelles. Its main function is to maintain homeostasis, which is a necessary process for cell survival [13]. Autophagy also plays a very important role in placenta formation [14,15]. E_2_ prolongs the survival of resting blastocysts by activating autophagy, and autophagy is very important for prolonging the life of resting blastocysts in utero during delayed implantation [16]. P_4_ can also activate the autophagy process of bovine mammary epithelial cells by inhibiting the mechanistic target of rapamycin (mTOR) pathway [17]. Stimulation and maintenance of endometrial function by P_4_ are essential for the development of pregnancy, implantation, placenta formation, and embryo development [18]. Abnormally low levels of P_4_ in the early luteal stage can cause embryo retardation in sheep and cattle [19]. The mTOR kinase is an indispensable part of the PI3K/AKT/mTOR signaling pathway and is considered to be a major negative regulator of autophagy, and the mTOR-specific inhibitor known as rapamycin can activate autophagy. Phosphorylation of mTOR is essential for cell-cycle progression, proliferation, and survival.

In early pregnancy and gestation in sheep, the uterus expresses prostaglandin I2 synthase (PTGIS), prostacyclin receptor (PTGIR), peroxisome proliferator-activated receptors (PPARs), and retinoic X receptors (RXRs) [11]. In the prostaglandin (PG) synthesis pathway, prostaglandin endoperoxide synthase 1 (PTGS1) and prostaglandin endoperoxide synthase 2 (PTGS2) catalyze the formation of prostaglandin H2 (PGH2) from arachidonic acid (AA), and PGH2 is converted into PGF_2α_ and prostaglandin E2 (PGE2) by prostaglandin F synthase (PGFS) and prostaglandin E synthase (PTGES), respectively [20].

Previous studies suggested that the activation of PPARγ was correlated with PGs synthesis, and PPARγ may affect the process of conceptus elongation and implantation by regulating the synthesis and secretion of PGs. PPARγ also shows crosstalk with the mTOR pathway, which may play a key role in mammalian embryo implantation. In this study, we tested the hypothesis that PPARγ/mTOR is involved in the regulation of PGs secretion in sheep trophoblast cells (STCs) during early pregnancy. These findings may provide new insights into the mechanisms by which hormones regulate the secretion of PGs in STCs during early pregnancy.

## 2. Materials and Methods

### 2.1. Ethics Statement

All procedures involving the collection of animal samples were approved by the Institutional Animal Care and Use Committee (IACUC) of the Shihezi University (A2020-149-01).

### 2.2. Cell Culture and Treatments

Six Kazakh ewes weighing between 45 and 55 kg and aged between 10 and 12 months were selected for this study. First, a plug was inserted in the vagina of each sheep and removed 12 days later. At the same time, pregnant horse serum gonadotropin (Solarbio, Beijing, China) was injected intramuscularly. After estrus induction, the same Kazakh ram was used to inseminate the ewes, and the last mating time was recorded as 0 d. The sheep were transferred to the slaughterhouse on day 16. Ovine uterine tissues were collected and placed in cold phosphate-buffered saline (PBS) (HyClone, South Logan, UT, USA) containing 500 U/mL penicillin and 500 ng/mL streptomycin, and then immediately transferred to the laboratory. The uterine tissue was washed with 70% ethanol to disinfect the surface and dissected on a sterile, ultraclean table. The uterine horn was rinsed with PBS containing 100 U/mL penicillin and 100 ng/mL streptomycin to obtain the conceptus, and the trophectoderm was placed in an EP tube and cut into pieces. The tissue was washed with PBS until the liquid was clear [21]. The chopped tissue was put into a 15 mL centrifuge tube, digested with 6 mL of 0.25% trypsin for 10 min, centrifuged at 1000× *g* for 10 min, and the supernatants were discarded. The samples were washed three times with PBS to remove pancreatic enzymes and transferred into 6-well plates containing 1 mL of DMEM with 10% fetal bovine serum (FBS) (Biological Industries, Kibbutz Beit Haemek, Israel), 100 U/mL penicillin, and 100 ng/mL streptomycin (HyClone, South Logan, UT, USA). The plates were incubated at 37 °C with 5% CO_2_. After 24 h incubation, 1 mL of fresh complete medium was added to the cultures. After 48–72 h, the cells were sub-cultured, and the expression of cytokeratin 7 (CK7) was determined using an SABC-AP kit with anti-rabbit IgG (IHC and ICC) (Beyotime, Shanghai, China) to determine the phenotype of the STCs. A subset of negative groups were stained for their nuclei using hematoxylin (Beyotime, Shanghai, China).

When the STCs reached 70–80% confluence, they were treated with 10-9 M estradiol (E_2_) (Sigma, St. Louis, MO, USA), 10-7 M progesterone (P_4_) (Sigma, St. Louis, MO, USA), and IFN-τ (20 ng/mL, Sangon Biotech Co., Ltd, shanghai, China) for 12 h [22]. After incubation, the culture supernatants and cells were collected by centrifugation. In the drug treatment group, GW9662 and rapamycin were added 2 h before hormone treatment, and samples were collected 12 h after hormone treatment. For the group treated with both GW9662 and rapamycin, GW9662 was first added for 2 h; then, rapamycin was added for 2 h and, finally, hormone treatment was carried out for 12 h.

### 2.3. RNA Extraction and Real-Time Quantitative PCR (RT-qPCR)

Total RNA was extracted from STCs using TRIzol reagent (Thermo Fisher Scientific, Waltham, MA, USA). RNA purity was determined by the A260/A280 ratio. The RNA was reverse-transcribed into cDNA using the PrimeScript™ RT reagent kit (TaKaRa, Tokyo, Japan), following the manufacturer’s protocol. Real-time quantitative PCR reactions were performed in 8-tube strips (Roche, Basel, Switzerland) with a reaction volume of 20 µL, consisting of 10 μL of SYBR Green (Roche, Basel, Switzerland), 7 μL of ddH_2_O, 100 ng of cDNA, and 1 μM target-specific forward and reverse primers. The reaction mixture was amplified in a LightCycler®96 (Roche, Basel, Switzerland) in three steps with a total of 40 cycles: 95 °C for 10 min, and then 40 cycles of 95 °C for 10 s, 60 °C for 1 min, and 72 °C for 1 min. Relative gene expression was calculated by the 2^-ΔΔCt^ method, and the mRNA expression level was normalized to the β-actin gene in all samples. All data were statistically analyzed using GraphPad Prism V. 8.0 (GraphPad Software, Inc., San Diego, CA, USA). Each sample was run in triplicate, and the experiments were repeated at least three times. The primers used for detection are shown in Table 1.

### 2.4. Western Blot Analysis

After the treatment of the STCs, total protein was extracted with RIPA lysis buffer (Cow Bio, Beijing, China), and the protein concentration was determined using a BCA kit (Beyotime, Shanghai, China). Protein samples were separated on an SDS-PAGE gel (BOSTER, Wuhan, China) and transferred to a 0.22 μm PVDF membrane by the semi-dry transfer method. After blocking in Tris-buffered saline containing 0.5% Tween-100 (TBST) and 10% nonfat dry milk for 1 h, the blots were incubated with anti-mTOR antibody (CST, Danvers, MA, USA), anti-phospho-mTOR (Ser2448) antibody (CST, Danvers, MA, USA), anti-p70S6K antibody (Beyotime, Shanghai, China), anti-phospho-p70S6K (Thr389) antibody (Beyotime, Shanghai, China), anti-BAX antibody (Beyotime, Shanghai, China), anti-LC3B antibody (Sigma-Aldrich, St. Louis, MO, USA), or anti-β-actin antibody (Beyotime, Shanghai, China) overnight at 4 °C. 

Subsequently, the membrane was washed three times with TBST for 10 min each time, and then incubated with anti-mouse DyLight™ 680 and anti-rabbit DyLight™ 800 (Abcam, San Francisco, CA, USA) secondary antibodies for 1 h at room temperature. After washing with TBST, the membrane was imaged with an Odyssey infrared imaging system (Li COR, Nebraska, NE, USA). The intensity values of the blotted protein bands were determined using ImageJ software (National Institutes of Health, Bethesda, MD, USA), and the relative expression levels of the proteins were calculated.

### 2.5. Enzyme-Linked Immunosorbent Assay (ELISA)

The STCs were inoculated in 96-well plates (2 × 10^3^ cells/well). After drug treatment, the cell supernatants were collected from each group and centrifuged at 4 °C for 10 min at 1000× *g*. The concentrations of PGF_2α_ and PGE_2_ in the culture medium were determined using an ELISA kit (J&L Biological, Shanghai, China), following the manufacturer’s instructions.

### 2.6. Statistical Analysis

All data were statistically analyzed using GraphPad Prism V. 8.0 (GraphPad Software, Inc., San Diego, CA, USA). The data were tested for normality, and they also demonstrated homogeneity of variance. One-way ANOVA followed by Tukey’s post hoc test and Fisher’s LSD test was used for multiple comparisons. Data for the study were derived from three separate repeated experiments and were expressed as the mean ± standard deviation (SD). Statistical differences were considered significant at *p* < 0.05.

## 3. Results

### 3.1. Morphological Characteristics of Sheep Trophoblast Cells (STCs)

Primary STCs isolated from sheep tissues have epithelioid growth and morphological diversity. With increasing passage number, trophoblast cell proliferation was significantly reduced. Growth stopped at the sixth passage and a large number of cells died due to aging (Figure 1A). The isolated cells expressed the trophoblast signature protein, CK7, as detected by immunohistochemistry (Figure 1B). In culture, intercellular fusion formed binucleated or multinucleated trophoblast cells, such as multinucleated syncytia (Figure 1C). More than 80% of the cultured cells were mononuclear.

### 3.2. Hormone Treatment of STCs Promoted PPARγ Expression, Inhibited the mTOR Pathway, and Increased Autophagy

To investigate the effects of hormone treatment on PPARγ protein expression and the mTOR pathway in STCs, we treated the STCs with hormones and observed protein changes through Western blotting. The results showed that, compared with the control group, hormone treatment significantly increased the expression of PPARγ protein while inhibiting the expression of p-p70S6K, p62, and LC3B-II/LC3B-I (Figure 2A–E). In addition, comparing the CON group with the E_2_+P_4_ group showed that IFN-τ treatment of the STCs increased the transcription level of ISG15 (Figure 2F). PPARγ expression increased to similar levels in both hormone treatment groups (Figure 2G).

### 3.3. Simulated Intrauterine Treatment of in STCs Increased the Release of PGs

To evaluate the effects of E_2_, P_4_, and IFN-τ on the levels of PGs in the STCs, we measured the transcription of PGs’ synthases and the concentrations of PGE_2_ and PGF_2α_ in the culture medium 12 h after hormone treatment. Real-time quantitative PCR showed that the E_2_+P_4_+IFN-τ group increased the mRNA levels of PTGS1 and PTGS2 in STCs but decreased them in the E_2_+P_4_ group. Compared with CON, the transcription levels of PTGES and PGFS in cells were lower after hormone treatment. However, compared with the E_2_+P_4_ group, the transcription of PTGES increased after the addition of IFN-τ, while the level of PGFS decreased (Figure 3A). ELISA results showed that PGE_2_ secretion increased under the E_2_+P_4_+IFN-τ treatment, while PGF_2α_ secretion was not affected. In contrast, PGE_2_ secretion did not change in the E_2_+P_4_ group, while PGF_2α_ secretion increased, which means that the ratio of PGE_2_/PGF_2α_ increased in the presence of IFN-τ (Figure 3B). 

### 3.4. Inhibition of PPARγ Activity Reduced the Release of PGs from STCs

To clarify the regulatory role of PPARγ in the secretion of PGs from STCs, we inhibited the activity of PPARγ in STCs by adding GW9662 2 h before hormone treatment. Compared to the E_2_+P_4_+IFN-τ group, the inhibition of PPARγ activity in the STCs significantly reduced the transcription of PTGS2 and PTGES but increased the transcription of PTGS1 and PGFS (Figure 4A). The corresponding ELISA results also showed that inhibition of PPARγ activity decreased the secretion of PGE_2_ and increased the secretion of PGF_2α_, resulting in a decrease in the ratio of PGE_2_/PGF_2α_ (Figure 4B).

### 3.5. Blockade of the mTOR Pathway Partially Inhibited the Normal Functional Release of Prostaglandins

To determine the role of the mTOR pathway in the secretion of PGs by STCs, we added rapamycin to block the pathway and measured its effects on PGs synthesis in the STCs. As shown in Figure 5A, compared with the E2+P4+IFN-τ group, blocking the mTOR pathway significantly reduced the transcription of PTGS2 and PGFS, while it increased the transcription of PTGS1 and PTGES. Rapamycin pretreatment increased the secretion of PGE_2_ and the ratio of PGE_2_ to PGF_2α_ but decreased the secretion of PGF_2α_ (Figure 5B).

### 3.6. Blocking the mTOR Pathway Can Restore the Functional Release of PGs in STCs after Inhibiting PPARγ Activity

To investigate the role of the mTOR pathway in PPARγ’s regulation of STCs’ secretion of PGs, we added GW9662 to inhibit PPARγ activity before rapamycin treatment. Real-time quantitative PCR results showed that after inhibiting PPARγ activity, adding rapamycin to block the mTOR pathway increased the transcription of PTGS1, PTGS2, and PTGES, while it inhibited PGFS transcription (Figure 6A). Notably, rapamycin pretreatment decreased the PGF2α secretion and increased the ratio of PGE_2_ to PGF_2α_. However, no increase in PGE_2_ was detectable in the rapamycin-pretreated group (Figure 6B).

### 3.7. Blockade of the mTOR Pathway Suppressed GW9662-Induced Inhibition of STCs’ Activity 

Western blot results showed that the inhibition of PPARγ activity increased the levels of p-p70S6k (Figure 7B)—a downstream marker of mTOR activity—and decreased the ratio of LC3B-II to LC3B-I (Figure 7C). Compared with the GW9662 group, blocking the mTOR pathway under conditions of inhibiting PPARγ activity could reduce the levels of p-p70S6k. Inhibition of PPARγ activity increased the expression of the apoptosis-related protein BAX, which was reduced by blocking the mTOR pathway (Figure 7D). 

## 4. Discussion

The placenta is an important barrier of maternal blood passing through the fetus. It is composed of a variety of cells, among which trophoblast cells are the most common. In this study, primary sheep trophoblast cells (STCs) were isolated by the tissue blocking method and trypsin digestion. An immunocytochemistry assay was performed on the isolated STCs, and the results showed that the cells expressed trophoblast-specific CK7 markers. Nuclear staining indicated that more than 80% of the cells were mononuclear, and these cells were used for the experiments. However, experimental studies with cell models cultured in vitro cannot truly be equivalent to studying the physiological state of trophoblast cells in the complex internal environment. Embryonic attachment is mainly co-regulated by E_2_, P_4_, and IFN-τ, and the results of this study showed that in the presence of E_2_, P_4_, and IFN-τ, the cell state was most similar to the physiological state at the embryo implantation stage. Interferon-stimulated gene-15 (ISG15) expression is induced by IFN-τ via paracrine and endocrine actions [23]; we found that ISG15 was not elevated in the presence of E_2_ and P_4_, indicating that the STCs did not secrete enough IFN-τ in vitro to produce an effect with E_2_ and P_4_ stimulation. In order to study the STCs as closely to the physiological state as possible, we used E_2_, P_4_, and IFN-τ together to stimulate STCs in vitro.

In early pregnancy, a series of prostaglandins are synthesized in the conceptus and uterus of sheep and cattle. The endometrial cavity in early pregnancy contains more prostaglandins than during the estrous cycle [24]. The trophectoderm of the conceptus mainly expresses PTGS2. Although the anti-lysolytic corpus effect of IFN-τ in pregnant cows arises from inhibition of the expression of endometrial epithelial and adrenal epithelial oxytocin receptors, it has not been found to inhibit the expression of PTGS2—the rate-limiting enzyme in prostaglandin synthesis [25]. Selective inhibition of PTGS in goats resulted in an inability to maintain the conceptus [26]. Prostaglandin receptors can be detected in all types of cells in the early pregnancy of sheep, and the prostaglandins regulated by PTGS2 may control endometrial function in early pregnancy through paracrine, autocrine, and cellular secretion [27]. When animal embryos undergo implantation, PGF_2α_ secretion is inhibited to preserve luteal function [28]; therefore, the ratio of PGE_2_ to PGF_2α_ is critical for embryo attachment. To investigate the molecular mechanisms regulating trophoblast cells, STCs were stimulated with exogenous P_4_, E_2_, and IFN-τ. We found that the ratio of PGE_2_ to PGF_2α_ decreased only in the presence of E_2_ and P_4_, while the ratio of PGE_2_ to PGF_2α_ increased when IFN-τ was added. This is consistent with the finding that the environment in utero is mainly regulated by E_2_, P_4_, and IFN-τ during embryo implantation. Gene expression levels related to peroxisome formation, fatty acid activation, and oxidation were upregulated during pregnancy lengthening in cattle [10]. After drug-induced upregulation of PPARγ in sheep during the implantation stage, the expression of implantation-related genes and key genes of the progenitor prolongation phase in trophoblast cells was determined to be upregulated to varying degrees [29]. Nutrients, including appropriate concentrations of the fatty acid ligand in the organization, can cause transcription factor activity and changes in cell biology, which may be necessary for the extension of the pregnant body. Insufficient PPARγ fatty acid ligands or imbalances in the organization of the nutrient composition of fatty acids may prevent implantation and lead to pregnancy failure [12]. Studies using human trophoblast cells in vitro have found that PPARγ can regulate the uptake of fatty acids by cells and induce the accumulation of lipids and trophoblast cell differentiation [30]. PPARγ is involved in prolongation of preimplantation embryos [31]. Similarly, the results of this study showed that the expression level of PPARγ protein was significantly increased in STCs under E_2_-, P_4_-, and IFN-τ-stimulated conditions. However, stimulation of STCs with E_2_ and P_4_ in vitro can increase PPARγ transcription, which is not affected by IFN-τ.

There are different targets and corresponding ligands in different structural domains of the PPARγ protein, among which arachidonic acid—which is the precursor for prostaglandin synthesis—and its metabolites are some of PPARγ’s natural endogenous ligands [32]. We noted that inhibition of PPARγ activity by GW9662 decreased the PGE_2_/PGF_2α_ ratio, increased apoptosis-related protein expression, and inhibited autophagy levels. This suggests that inhibition of PPARγ activity is detrimental to STCs.

Inhibition of PPARγ activity can enhance mTOR signaling [33]. The mTOR signaling system regulates a wide range of cellular processes and plays an important role in fertilization, embryogenesis, placenta formation, and preimplantation embryo development [34]. The downstream target of mTOR, p70S6K, is phosphorylated by mTOR to negatively regulate the synthesis of proteins involved in autophagy [35]. The phosphorylation level of p70S6K can be used to reflect mTOR activity. LC3-II, which is expressed in most cell types, is a structural component of mature autophagosomes; thus, it is often used as a specific marker of autophagy [36]. The autophagy-associated protein p62 is inversely associated with autophagy activity [37]. Previous studies have shown that the mTOR signaling pathway is important in early embryonic development and embryonic stem cell proliferation [38]. During normal embryonic attachment, the mTOR pathway’s activity is inhibited. In the present study, we found that mTOR activity was reduced and the autophagy level was increased after stimulation of STCs with E_2_, P_4_, and IFN-τ, and that blocking the mTOR pathway could further increase the ratio of PGE_2_ to PGF_2α_.

Overexpression of PPARγ inhibited the activation of the AKT/mTOR/p70S6K signaling cascade [39]. Rapamycin can inhibit the proliferation and differentiation of 3T3-L1 and human precursor adipocytes by inhibiting the activity of PPARγ [40]. In the regulation of lipid synthesis, mTOR upregulated PPARγ activity to promote the synthesis and deposition of lipids [41]. Therefore, it is possible that the epistatic relationship between the two molecules is dependent on the environment, or that PPARγ is activated by mTOR and participates in a negative feedback loop to limit mTOR activation [42]. In order to further confirm the role of PPARγ/mTOR in early pregnancy, we used rapamycin to block the mTOR pathway after inhibiting PPARγ activity. The results showed that, under the conditions of inhibiting PPARγ activity, blocking the mTOR pathway increased the ratio of PGE_2_ to PGF_2α_, inhibited autophagy, and reduced apoptosis. These results suggest a plausible role for PPARγ/mTOR in regulating PG secretion. More detailed molecular mechanisms require further studies.

## 5. Conclusions

Our findings suggest that hormones upregulate the expression of PPARγ in STCs and increase the ratio of PGE_2_/PGF_2α_ by increasing PGE_2_ secretion. Inhibition of PPARγ activity under hormone treatment reversed the increase in the PGE_2_/PGF_2α_ ratio. However, blocking the mTOR pathway under hormone treatment can restore this level, further confirming that PPARγ/mTOR are involved in the hormonal regulation of PGs secretion. These findings may provide new insights into the mechanisms by which hormones regulate PGs secretion in STCs and the biological functions of PPARγ and mTOR.

## Figures and Tables

**Figure 1 vetsci-09-00649-f001:**
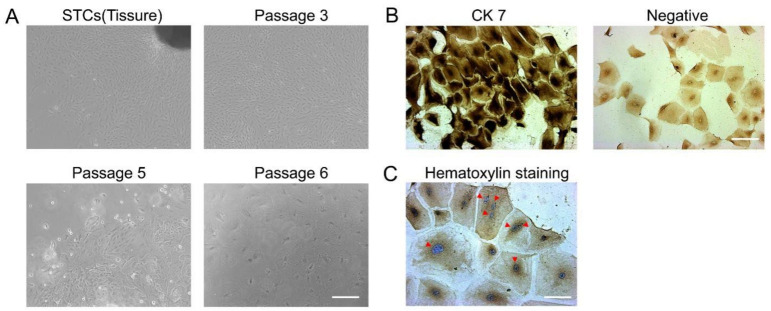
Isolation, culture, and identification of primary STCs: (**A**) Primary cultured STCs were observed under a light microscope (scale bar, 200 μm). (**B**) Immunohistochemistry of cytokeratin CK7 (scale bar, 100 μm). (**C**) Blue nuclei can be seen after hematoxylin staining (scale bar, 50 μm). Red arrows indicate nuclei.

**Figure 2 vetsci-09-00649-f002:**
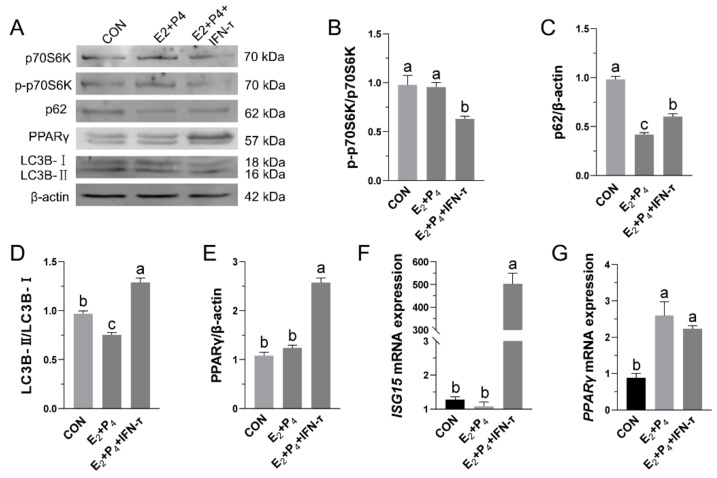
Effects of hormone treatment on PPARγ and mTOR pathway proteins: (**A**–**E**) After 12 h of hormone treatment, the expression of p70S6K, p-p70S6K, p62, LC3B-I, LC3B-II and PPARγ in STCs was determined by Western blotting (Appendix A). Real-time quantitative PCR analysis of (**F**) ISG15 and (**G**) PPARγ in STCs. The β-actin gene was used as a normalization control. In the CON group, 0.1% DMSO was added. Data are shown as the mean ± standard deviation (SD) of three independent trials. Different letters above the bars indicate significant differences (*p* < 0.05).

**Figure 3 vetsci-09-00649-f003:**
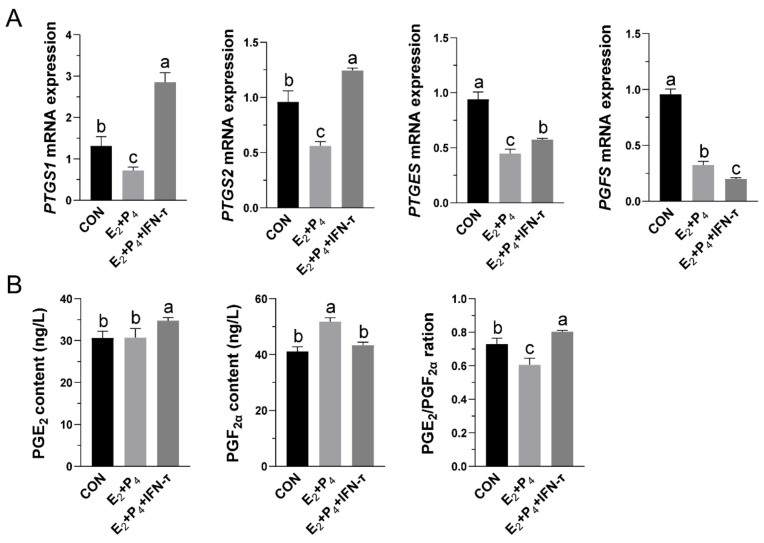
Effects of hormones on the secretion of PGs in STCs: (**A**) After 12 h of hormone treatment, the relative mRNA level of prostaglandin synthase was determined by real-time fluorescent quantitative PCR. The β-actin gene was used for normalization in all samples. (**B**) Secreted PGE_2_ and PGF_2α_ were measured by ELISA after 12 h of hormone treatment, and the ratio of PGE_2_/PGF_2α_ was calculated. Data are shown as the mean ± standard deviation (SD) of three independent trials. In the CON group, 0.1% DMSO was added. Different letters above the bars indicate significant differences (*p* < 0.05).

**Figure 4 vetsci-09-00649-f004:**
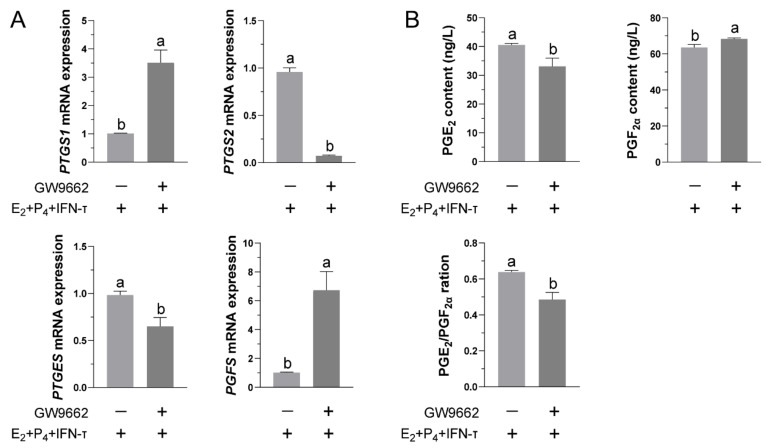
Effects of the inhibition of PPARγ activity in STCs on the secretion of PGs after hormone treatment: (**A**) After inhibiting PPARγ activity with GW9662, the relative mRNA levels of prostaglandin synthases in STCs were measured by real-time quantitative PCR. The β-actin gene was used as a control gene. (**B**) The concentrations of PGE_2_ and PGF_2α_ were determined by ELISA after inhibiting PPARγ activity, and the ratio of PGE_2_/PGF_2α_ was calculated. Data are shown as the mean ± standard deviation (SD) of three independent trials. Different letters above the bars indicate significant differences (*p* < 0.05).

**Figure 5 vetsci-09-00649-f005:**
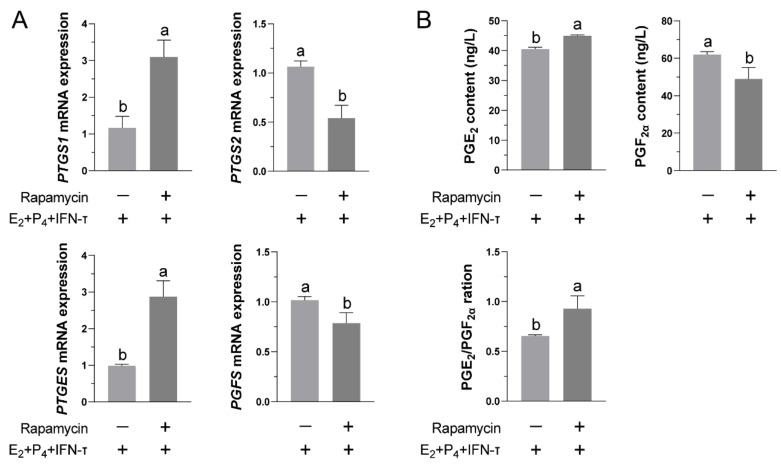
Effects of blocking the mTOR pathway on the secretion of PGs by STCs after hormone treatment: (**A**) Relative mRNA levels of prostaglandin synthases in rapamycin (Rap)-pretreated STCs were determined by real-time fluorescence quantification. The β-actin gene was used as the control in all samples. (**B**) After blocking the mTOR pathway, the secretion of PGE_2_ and PGF_2α_ was measured by ELISA, and the ratio of PGE_2_/PGF_2α_ was calculated. Data are shown as the mean ± standard deviation (SD) of three independent trials. Different letters above the bars indicate significant differences (*p* < 0.05).

**Figure 6 vetsci-09-00649-f006:**
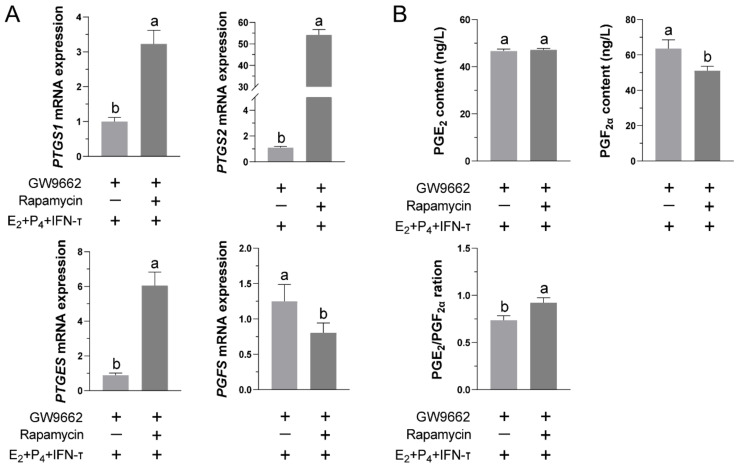
Influence of crosstalk between PPARγ and mTOR on the secretion of PGs in STCs under hormone treatment: Under hormone treatment, the mTOR pathway was first blocked, and then GW9662 was added for preconditioning. The culture medium supernatants were collected, and PGs were measured by ELISA, while RT-qPCR of the total RNA of the STCs was performed to determine the mRNA levels. (**A**) The relative mRNA level of the rate-limiting enzyme in PGs synthesis was quantified by RT-qPCR, with the β-actin gene used as the control in all samples. (**B**) Secretion of PGE_2_ and PGF_2α_ was detected by ELISA, and the ratio of PGE_2_/PGF_2α_ was calculated. Data are expressed as the mean ± SD of three independent experiments. Different letters above the bars indicate significant differences (*p* < 0.05).

**Figure 7 vetsci-09-00649-f007:**
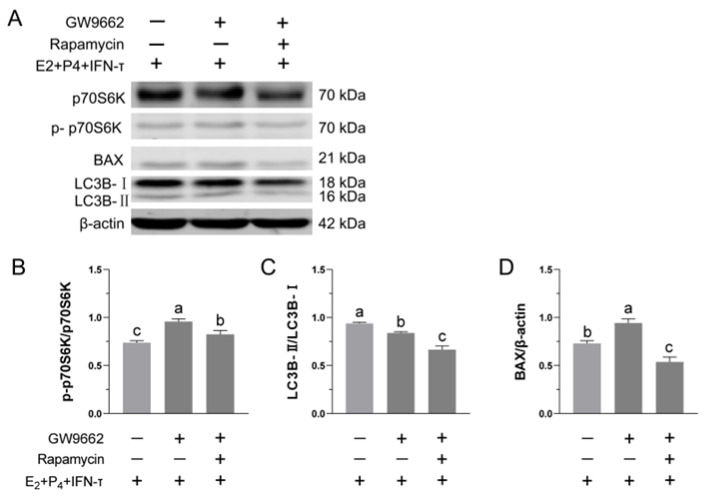
Blocking the mTOR pathway can alleviate the adverse effects of PPARγ inhibition: (**A**) Protein levels of p70S6K, p-p70S6k, LC3B-I, LC3B-II and BAX were determined by Western blotting (Appendix A). ImageJ grayscale intensity measurements were performed on protein bands, and the ratios of (**B**) p-p70S6K/p70S6K, (**C**) LC3B-II/LC3B-I, and (**D**) BAX/β-actin were calculated. Data are expressed as the mean ± SD of three independent trials. Different letters above the bars indicate significant differences (*p* < 0.05).

**Table 1 vetsci-09-00649-t001:** Primers used for qPCR determination of relative mRNA expression.

Gene Name	GenBank	Sequence (5’–3’)	Length (bp)
PTGS1	NM_001009476.1	F: CATCCACTTTCTGCTGACGC R: GGATAAGGTTGGAACGCACTG	110
PTGS2	NM_001009432.1	F: ATCCCCAGGGCACAAATCT R: TTGAAAAGGCGACGGTTATG	175
PTGES	XM_027966307.2	F: AGTCCTGGAGCTAATGAACGG R: TTCTTCCGCAGCCTCACTT	117
PGFS	XM_004014323.5	F: CAGTTCTTTGTGCCATTGCC R:CTCTTTGATCCGCTTCTTGTTG	122
β-actin	U39357.2	F: AGGTCATCACCATCGGCAAT R: CGTGTTGGCGTAGAGGTCTTT	155
ISG15	NM_001009735.1	F: TGAAGGTGAAGATGCTAGGGG R: GCTGGAAAGCAGGCACATT	114
PPARγ	KF727439.1	F: CTCATAATGCCATCAGGTTCG R: CAGCAGACTCTGGGTTCAGTTG	102

## Data Availability

Not applicable.

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
