# Peer review of "PPARγ/mTOR Regulates the Synthesis and Release of Prostaglandins in Ovine Trophoblast Cells in Early Pregnancy"

_vetsci, 2022, doi:10.3390/vetsci9110649_

Round 1

Reviewer 1 Report

Thank you for the opportunity to review the manuscript entitled "PPARγ/mTOR regulates prostaglandin synthesis and release in ovine trophoblast cells in early pregnancy". This manuscript details an in vitro study of ovine trophoblast prostaglandin regulation. The data is presented reasonably well, however, there are some major concerns that are detailed below. 

Introduction

Line 31- To the best of my knowledge "infixed" is not a word. Attached or affixed would be better. 

Line 36/44/51/67/87 etc- ovine embryos do no "implant". Please revise throughout the manuscript. 

No hypothesis given. 

Material & Methods

-How many sheep were used to derive tissue. Were they of a similar genetic background? Were multiple sires used?

- Was any attempt made to differentiate binuclear and mononuclear prior to culture and passage?

-It is unclear when samples were obtained after hormone treatment and for how long the cells had been exposed. 

-Please indicate the method used for comparison between transcript levels. 

-Please indicate tests used for detection of normality and post-hoc group comparisons. 

Results

Line 168- I believe you mean mononuclear. Can "mainly" be quantified as this may be a biologically relevant feature not controlled for in this study.

Line 188-189- Figure reference 2F and 2G do not match figure 2. Please revise. 

Control Groups- As the 'control' group frequently changes it may be best to use the group (i.e. E+P+IFNt) on figures to increase clarity as to which groups are actually being compared. 

Line 192- please rephrase to in vitro. This is not a simulated intrauterine environment. That would minimally require co-culture with endometrial cells. Pleas amend elsewhere in the text if used. 

Line 275- this experiment is not "pregnancy" please amend as it is misleading. 

Discussion

Line 301- This is not simulated pregnancy, attachment etc. Please revise throughout. 

-Please discuss the limitations of this study. This may include the inherent limitations of cell culture/n vitro  models, the nature of the treatment groups (i.e. why only combinations and not hormones alone etc), suitability of techniques used for assessment of nuclear receptor activity etc. 

Reviewer 2 Report

The article discusses about "determine whether PPARγ/mTOR was involved in the regulation of PG secretion in sheep trophoblast cells (STCs) during early pregnancy". However, I consider that it can be published if some adjustments are made.  Reconsider after major corrections.

Simple Summary: Add to manuscript

Abstract:

Line 17 Correct "prostaglandin-E2 ratio" to "prostaglandin-E2 ratio".

Line 18 Please mention the meaning of the acronyms

Materials and Methods

General: describe the experimental design and the groups into which they were divided, number of samples, how the animals were handled, time, age, whether tissue samples were biopsied or sacrificed, how gestation was synchronized or how gestation time was determined, etc.

Specifics:

Line 160.- It is necessary to expand the "Statistical Analysis" section. The data were tested for normality, and also had homogeneity of variances. It is necessary to describe the analyses for the variables.

Results

General: Increase the size of the literals of the graphs, they are too small to be read.

Specific:

Line 183.- change "in vitro" to "in vitro"?

Line 203.- revise "E2,P4" is not "E2+P4"?

Line 237-239.- This reference "The mTOR signaling system regulates a wide range of cellular processes and plays 237 an important role in embryogenesis, normal placentation, fertilization, and preimplantation embryonic 238 development [25]." It goes in the discussion section. Only results are reported here.

Figure 6. in the graph where the PTGS2 mRNA expression results are presented the sub index "a" should be in the GW+Rap bar.

Discussion

General: I recommend restructuring this section. That is, adjust this section according to how I report the results. I recommend discussing concerning what could have been the cause of the findings found, expand this section.

I specify:

Line 291.- I suggest, if your results start talking about "Morphological characteristics of sheep trophoblast cells" it would be convenient to start with this section.

Line 304-305.- continue explaining "we found that the ratio of PGE2 to PGF2α decreased only in the presence of E2 and P4, while the ratio of PGE2 to PGF2α increased when IFN-τ was added" to what can we attribute these results? discuss.

Line 323.- Explain the relationship of mTOR and PPARγ with respect to "p-p70s6k".

Conclusion:

General: expand the discussion to better support the conclusion section. That is, better describe the pathways involved in each process.

References

In the references there are citations that are not in accordance with the format of the journal, for example 6,7,8,10... the name of the journal should be abbreviated, please correct it.

Round 2

Reviewer 2 Report

The manuscript has improved considerably, I suggest its publication in the present form.